# Measurement of Multi Ion Transport through Human Bronchial Epithelial Cell Line Provides an Insight into the Mechanism of Defective Water Transport in Cystic Fibrosis

**DOI:** 10.3390/membranes10030043

**Published:** 2020-03-12

**Authors:** Miroslaw Zajac, Andrzej Lewenstam, Piotr Bednarczyk, Krzysztof Dolowy

**Affiliations:** 1Institute of Biology, Department of Physics and Biophysics, Warsaw University of Life Sciences—SGGW, 159 Nowoursynowska St., 02-776 Warsaw, Poland; Miroslaw_Zajac@sggw.pl (M.Z.); piotr_bednarczyk@sggw.pl (P.B.); 2Faculty of Materials Science and Ceramics, AGH University of Science and Technology, Mickiewicza 30, 30-059 Krakow, Poland; andrzej.lewenstam@gmail.com; 3Faculty of Chemistry, Biological and Chemical Research Centre, University of Warsaw, Żwirki i Wigury 101, 02-089 Warsaw, Poland

**Keywords:** ion transport, water transport, epithelium, cystic fibrosis

## Abstract

We measured concentration changes of sodium, potassium, chloride ions, pH and the transepithelial potential difference by means of ion-selective electrodes, which were placed on both sides of a human bronchial epithelial 16HBE14σ cell line grown on a porous support in the presence of ion channel blockers. We found that, in the isosmotic transepithelial concentration gradient of either sodium or chloride ions, there is an electroneutral transport of the isosmotic solution of sodium chloride in both directions across the cell monolayer. The transepithelial potential difference is below 3 mV. Potassium and pH change plays a minor role in ion transport. Based on our measurements, we hypothesize that in a healthy bronchial epithelium, there is a dynamic balance between water absorption and secretion. Water absorption is caused by the action of two exchangers, Na/H and Cl/HCO_3_, secreting weakly dissociated carbonic acid in exchange for well dissociated NaCl and water. The water secretion phase is triggered by an apical low volume-dependent factor opening the Cystic Fibrosis Transmembrane Regulator CFTR channel and secreting anions that are accompanied by paracellular sodium and water transport.

## 1. Introduction

Epithelial tissue is made of a single layer of cells bound together by tight junctions. It forms a barrier to the simple diffusion of water and ions. Water is transported across the epithelial cell layer by means of osmosis. Electroneutral transport of a single cation accompanied by a single anion causes the passive osmotic flow of 370 water molecules across the epithelium. In cystic fibrosis (CF), the anion transport route is impaired, which leads to defective ion and water transport across the epithelium and a too dense secreted mucus.

Epithelial cells are polarized, i.e., there are different transporting molecules on both the apical and basolateral surfaces of the epithelial cells. There is a considerable number of different ion channels, transporters and pumps molecules that are present in epithelial cell membranes [1,2,3,4,5,6,7].

Despite half a century of research, our knowledge of the interdependence of the ion fluxes and water transport across the epithelial cell layer is limited for several reasons. Firstly, there are many ion channels, transporters and pumps that are involved in transport. Secondly, there are at least five ions transported simultaneously in both directions across the epithelial cell layer i.e., chloride, bicarbonate, sodium, potassium and hydrogen. Thirdly, a method is lacking that would allow for simultaneous measurements of all ions transported within very short time intervals (second-time range). A review of electrochemical and radiotracer methods of ion transport across the epithelium was published recently [8]. There are continuous attempts to increase the number of methods to study epithelial transport e.g., the fluorescence method has been employed [9]. However, these contributions do not provide an explanatory vision of the interdependence of ion fluxes and dense mucus formation.

After a decade of tests and experiments, we succeeded in constructing a device that measures the transport of four ions (sodium, potassium, chloride and pH) across the epithelial cell monolayer grown on porous support by means of ion-selective electrodes [10,11,12]. Before we started our experiments, it was generally accepted that water moves across the epithelial layer due to the osmotic difference between the apical and basolateral face. Thus, we used two isosmotic solutions that differed with either sodium or chloride concentrations. It was not clear whether a single cell line could transport ions in both directions or only in one. We used gradients with higher sodium or chloride concentrations either on the apical or on the basolateral sides. It was generally believed that ions are transported through the cell i.e., via a transcellular route. Therefore, we used blockers of sodium and chloride channels and transporters in high concentrations to try to prevent transcellular ion transport. The results of our experiments shown in this paper suggest that the generally accepted views in the field of epithelial transport are often wrong.

The results obtained using this method allow offering a more consistent mechanism of apical surface liquid (ASL) equilibrium than any other offered so far. We could now test the hypothesis proposed herein of a dynamic water balance across the human bronchial epithelium.

## 2. Materials and Methods

### 2.1. Apparatus

The measuring system has been described earlier [11,12] and is presented in Figure 1. Briefly, the Costar Snapwell inserts with epithelial cell monolayer are sandwiched between two asymmetric probes containing four ion-selective electrodes (Na, K, pH and Cl), the reference electrode, inlet and outlet tubes. The measuring platform is flat and placed 20–40 µm from the surface of the cell layer. The apical and basolateral chamber volumes are similar and equal to approximately 30 μL i.e., only 2–3 times the volume of the cell layer. The electromotive forces (EMFs) of the ion-selective electrodes were measured versus the reference electrode by means of a 16-channel Lawson Lab EMF interface connected to a PC with appropriate data acquisition software. A syringe pump (SP 260PZ, WPI) was used to exchange the media. We kept the speed of media exchange at 0.3 mL/min to avoid damage to the cell layer. The stability of the cell layer was monitored by the measurement of the transepithelial resistance. We observed a dyed medium exchange under the microscope using a modified Costar Snapwells, where the porous membrane was replaced by a glass slide. We found that the medium flow is laminar. After a few chamber volumes had passed, the color of the medium was uniformly distributed within the chamber, excluding the possibility of the medium composition difference parallel to the cell plane. In our experimental conditions, the complete medium exchange took 24 s. Measurements were made after medium flow stopped. The small thickness of the medium layer excluded the possibility of the unstirred layer effect since the ion concentration differences decay within less than one second. The equilibration of our ISE electrodes took 30–45 s as shown in Figure 2. Both tube outlets were open and placed at the same heights to avoid hydrostatic pressure formation. All experiments were performed at room temperature.

### 2.2. Cells

The human bronchial epithelium 16HBE14σ cell line obtained from the late Dr. Dieter Gruenert University of California in San Francisco was grown as described earlier [11,12]. The cells were grown submerged in the culture medium for the first six to nine days. The medium from the apical side was then aspired. Only the basolateral (bottom) side of the cell monolayer was in contact with the medium, while the apical (upper) side was in contact with air supplemented with 5% carbon dioxide. The inserts were used for the experiment after 10–16 days after the air contact was established. The resistance of the cell monolayer used in the experiment was higher than 300 Ω∙cm^2^. The 16HBEσ cell line is widely used to study respiratory ion transport and the function of the CFTR channel. It is used as the best model of epithelial barrier functions. Additionally, the 16HBEσ cell line model was well suited for our measurements.

### 2.3. The Media

To study the effect of sodium and chloride concentration gradient across the epithelial cell layer, Krebs–Henseleit solution (KHS1) and its two modifications KHS2 and KHS3 were used (Table 1). All three solutions had the same ionic strength, ionic activity coefficient, pH and osmotic pressure.

To estimate the role of ion channels, transporters and pumps in the transport process, blockers of low specificity and many target molecules in very high doses were used. In the experiments, the KHS solutions were supplemented with 100 μM of amiloride; the potent apical epithelial sodium channel (ENaC) blocker [13] and the blocker of Na/H exchanger, and virtually all sodium-dependent processes; 100 μM of glibenclamide, the apical CFTR channel blocker [14] of K_ATP_ channel and VSOR channel [15] and other exchangers regulated by the ABC binding protein, 100 μM of 4,4′-Diisothiocyano-2,2′-stilbenedisulfonic acid (DIDS), the blocker of the ORCC channel [16,17] and other anion channels and exchangers [18].

The following additives to the media were used: step (I), no additives; (II), amiloride on the apical face of the epithelial cell monolayer; (III), as in step (II) plus glibenclamide and DIDS on both faces of the cell layer.

### 2.4. Electrodes

The method of fabrication of ion-selective electrodes has been described previously [11]. Each of the electrodes was calibrated separately before mounting them on the measuring platform. The calibration was determined by changing the appropriate ion concentration in the medium. The electrodes showed a linear relationship between the potential and log of ion activity. The slope for the chloride-sensitive electrode was equal to s = −58 ± 2 mV/dec., for the potassium-sensitive electrode +55 ± 1 mV/dec., for the sodium +54 ± 1 mV/dec. and for the hydrogen +57 ± 2 mV/dec. The potential of the ion-selective electrodes was measured against the commercial macro-reference Ag/AgCl electrode (InLab, Mettler Toledo). The micro-reference electrodes used in our system differ from the commercial one by ± 1 mV.

After changing the medium, the ISE potential changes and reaches 95% of the final potential value in 5 min. The slope for the chloride-sensitive electrode was equal to s = −57 ± 2 mV/dec. in KHS1 solution, s = −58 ± 2 mV/dec. in KHS2 and s = −40 ± 2 mV/dec. in KHS3. For the sodium-selective electrode the slope was equal to s = +51 ± 2 mV/dec. in KHS1 solution, s = +40 ± 2 mV/dec. in KHS2 and s = +38 ± 2 mV/dec. in KHS3 solutions.

### 2.5. Measurement of Multiple Ion Transport Procedure

The epithelial cells grown on porous support were mounted on the measuring platform. The KHS1 solution was introduced to both sides of the cell monolayer. After 10 min on one side of the cell layer (apical or basolateral), KHS2 or KHS3 solution was introduced. The changes in ISE electrode potentials were recorded for 10 min. In the next experimental step, the same solutions supplemented with blockers were introduced to the measuring chambers and the ISE potentials. Measurements were recorded for another 10 min after the medium exchange stop.

## 3. Results

We performed 269 single determinations measured in 51 series for four different pairs of sodium or chloride concentration gradients using 80 inserts. The results for sodium, chloride and potassium transport are expressed as the change of ion concentration on a particular side of the epithelial layer. Final pH and transepithelial potential difference measured against the potential at basolateral medium 10 min after the media change terminates; for four different media gradients are presented in Figure 3, Figure 4, Figure 5 and Figure 6. In all cases presented, the concentration gradient induces transport of both sodium and chloride ions from the side of higher to the side of lower concentration. Potassium transport plays a minor role in our experimental conditions. The final pH value is 7.15 ± 0.15 for all media combinations, which means that there is no considerable transport of bicarbonate or proton ions, or they are transported concurrently. It is likely that the pH is controlled in bronchial epithelial cells by at least three processes Na/H exchange, Cl/HCO_3_ exchange and by CO_2_/HCO_3_ equilibrium in which CO_2_ production, intracellular and extracellular carbonic anhydrases are involved [7,19]. The transepithelial potential value is less than 3 mV.

## 4. Discussion

The data presented in Figure 3, Figure 4, Figure 5 and Figure 6 show that the measured transport of cations is not equal to anions i.e., the transport is apparently not electroneutral. The transport of ions to the chamber with lower concentrations of chloride and sodium is of one order of magnitude higher than the decrement on the higher concentration side. The simple explanation that the export or import of bicarbonate anion provides the balance does not hold since bicarbonate transport should also affect the pH and the required amount of bicarbonate surpasses the amount available in the chambers.

Thus, another explanation for the apparent breaking of the electroneutrality and mass transport principle is required. We recalculated the concentration changes to the final concentration values which show that the concentration of the anions is equal to cation concentration—thus the electroneutrality principle is not violated. This allows us to assume that the change in concentrations of ions is caused by isosmotic transport of 145 mM NaCl solution across the cell layer and that the transported solution consists only of *x* fraction of the chamber volume. From this assumption, one can calculate the expected final concentration of particular ion C_fin_ starting from the initial concentration C_init_ as $145x+C_{\mathrm{init}}*\left(1-x\right)=C_{\mathrm{fin}}$ for the side to which NaCl solution is transported and $C_{\mathrm{init}}*\left(1+x\right)-145x=C_{\mathrm{fin}}$ for the side from which NaCl solution is transported. The volume of transported fluid (*x*) is in control conditions, approximately 25% of the chamber volume (equivalent to 5–10 μL) and less for the solutions supplemented with (Table 2). Using *x* values from Table 2, one obtains an almost perfect fit between the theoretical and the measured values of final sodium and chloride concentrations (Figure 7 and Figure 8).

We found that the imbalance of sodium or chloride ions concentrations on both sides of the epithelial cell monolayer causes the flux of isosmotic NaCl solution across the epithelial layer and thus decreasing the ion gradient. We found that the transepithelial resistance of the cell monolayer is not affected during our experimental procedure, which rules out the formation of the permanent fluid transport route across the cell layer. The results presented in Figure 7 and Figure 8 and Table 2 suggest that the blockers of chloride ion channels and transporters considerably affect the volume of fluid transported, while the blockers of sodium channels and transporters are less effective. This is in accordance with earlier finding that sodium is transported via a paracellular way while chloride is transported via a transcellular one [11]. According to Table 2, during 600 s of the experiment, 25% of 30 μL of chamber volume is transported across the 1.1 cm^2^ cell layer in the form of 145 mM NaCl solution; this is equivalent to the movement of 10^15^ ions per 1.1 cm^2^ per second or 1.5 pA/μm^2^ which can be delivered by a few open CFTR channels per μm^2^ or a single VSOR or ORCC channels per few μm^2^. However, further studies of the mechanism of fluid transport are necessary e.g., on whether the water is transported transcellularly via aquaporins [20] or paracellularly. While our experiments considered drastic nonequilibrium conditions, similar mechanisms are likely to be involved in near-equilibrium conditions in the lungs.

There are a few hypotheses of how epithelial cells regulate the thickness of apical fluid in vivo. Boucher [21,22] suggests that the same cells alternate between fluid secretion and absorption. Quinton [23] claims there are separate cells that independently secrete and absorb fluid. There is strong evidence that bicarbonate ions are transported across epithelial cells and are responsible for proper mucus consistency [24,25]. Considering our experimental results presented herein show that: (i) water is transported in both directions, (ii) bicarbonate and proton ions are transported concurrently, and (iii) our earlier findings that sodium ions are transported paracellularly while chloride ions are transported transcellularly [11], we propose ‘the dynamic water balance’ hypothesis and relevant model. The essence of our concept is presented in Figure 9. We assume that in a healthy bronchial epithelium, there is a dynamic balance between water absorption (Figure 9A) and water secretion (Figure 9B). The action of two exchangers, Na/H and Cl/HCO_3_, is responsible for the water absorption phase. In effect, with the ion transport conducted by these two exchangers, there is a flux of sodium and chloride into the cytoplasm and secretion of bicarbonate and proton ions from the cytoplasm to apical fluid (ASL). While sodium chloride is fully dissociated, the bicarbonate and proton form weakly dissociated carbonic acid, decreasing osmotic pressure. The lower osmotic pressure of ASL causes water transport into the cytoplasm; also, the carbonic acid present in ASL fluid slowly decomposes due to the presence of a small amount of carbonic anhydrase in the apical fluid [26]. The observation of Alexandrou and Walters [27] that the absorption of lung fluid is amiloride-sensitive (acting on Na/H exchanger) but not sensitive to chloride channels blocker is in accordance with our model. The exchange of chloride for bicarbonate results in an increase of bicarbonate concentration in the apical fluid and chloride concentration in the cytoplasm (and a decrease on the opposite sides). The energy gain of chloride for bicarbonate exchange is not far from electrochemical equilibrium. The small change of concentration of bicarbonate ion in the cytoplasm by only 5 mM and the increase of 5 mM in the ASL fluid (or 20 mM of chloride concentration change) would stop net ion transport through the Cl/HCO_3_ exchanger, while the Na/H exchanger will still operate leading to the acidification of ASL fluid.

During the absorptive phase, the volume of ASL decreases. We assume that an ASL volume-dependent factor i.e., ATP concentration in ASL as proposed earlier [28,29], triggers the opening of the CFTR channel. The opening of the CFTR channel starts the secretion phase (Figure 9B). The secretion of chloride and bicarbonate ions through the CFTR channel is accompanied by paracellular transport of sodium ions and the action of the Na/H exchanger. Taking the ion selectivity ratio of k_Cl/HCO3_ = 4 [30,31] for the CFTR channel, one can easily estimate that 2.6 chloride ion is secreted to ASL for 1 bicarbonate:
$$\frac{I_{\mathrm{Cl}}}{I_{\mathrm{HCO}3}}=\frac{k_{\mathrm{Cl}/\mathrm{HCO}3}\times \left(\frac{\mathrm{RT}}{F}\ln\left(\frac{\left[{\mathrm{Cl}}_{c}\right]}{{\mathrm{Cl}}_{\mathrm{ASL}}}\right)+\;\left(\varphi_{\mathrm{ASL}}-\varphi_{c}\right)\right)}{\frac{\mathrm{RT}}{F}\times \ln\left(\frac{\left[\mathrm{HCO}3_{c}\right]}{\left[\mathrm{HCO}3_{\mathrm{ASL}}\right]}\right)+\left(\varphi_{\mathrm{ASL}}-{\;\varphi }_{c}\right)}=2.6$$
where subscript ‘ASL’ denotes apical fluid and subscript ‘c’ cytoplasm; during the secretory phase, the net 2.6 molecules of NaCl appear on the apical side of the bronchial epithelium. The osmotic pressure of ASL increases, leading to water secretion to ASL. Water secretion dilutes the trigger molecule responsible for opening the CFTR channel. The CFTR channel closes, and the system switches back to the absorptive phase. In the absence of a functional CFTR channel, the trigger sensor is missing, and the secretory phase is no longer self-regulated by the end of the absorptive phase. The consequence of the lack of a functional CFTR channel is the low volume of ASL, dense mucus layer, and the acidification of ASL characteristic of CF. In this way, the hypothesis presented combines ion fluxes with water transport and explains the formation of ASL mucus in the case of dysfunctional CFTR.

The secretion of chloride into ASL is possible also via other anionic channels present in the apical face of the bronchial epithelium e.g., by the calcium-dependent anionic channel CaCC modeled recently [32]. However, the secretion through the CaCC channel (TMEM16A) due to the lower selectivity ratio k_Cl/HCO3_ = 2 [33], or even less [34], would lead to secretion of only 1.3 chloride per 1 bicarbonate ion and consequently, only half of the amount of water is secreted compared to a functional CFTR channel. Moreover, the system is not self-regulated.

## 5. Conclusion

In the isosmotic transepithelial concentration gradient of either sodium or chloride ions, there is an electroneutral transport of the isosmotic solution of sodium chloride in both directions across the cell monolayer.

In a healthy bronchial epithelium, there is a dynamic balance between water absorption and secretion. Water absorption is caused by the action of two exchangers, Na/H and Cl/HCO_3_, secreting weakly dissociated carbonic acid in exchange for well dissociated NaCl and water. The water secretion phase is triggered by an apical low volume-dependent factor opening the CFTR channel and secreting anions that are accompanied by paracellular sodium and water transport.

## Figures and Tables

**Figure 1 membranes-10-00043-f001:**
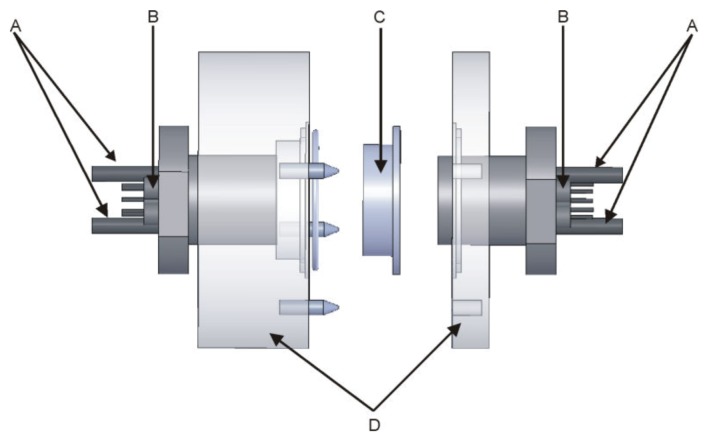
Schematic model of apparatus included. (**A**) solution inlet and outlet tubes, (**B**) ion-selective and reference electrodes, (**C**) Costar Snapwell insert with epithelial cell monolayer, and (**D**) asymmetric body. The diameter of the body is 4 cm. Distance between the cell layer and ion-selective electrodes is less than 30 µm. The inner diameter of the Costar Snapwell insert is 12 mm.

**Figure 2 membranes-10-00043-f002:**
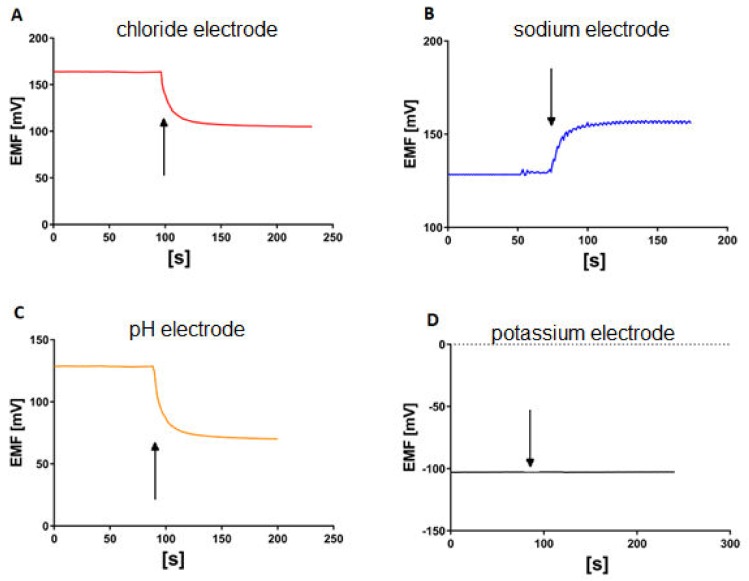
Kinetics of the ion selective electrodes response after changing the medium. The start of medium exchange is shown by the arrow. (**A**) KHS3 solution is replaced by KHS1, (**B**) KHS2 solution is replaced by KHS1, (**C**) KHS1 solution equilibrated to pH = 6.4 is replaced by KHS1 (pH = 7.4), (**D**) KHS3 solution is replaced by KHS1 (no change in potassium concentration).

**Figure 3 membranes-10-00043-f003:**
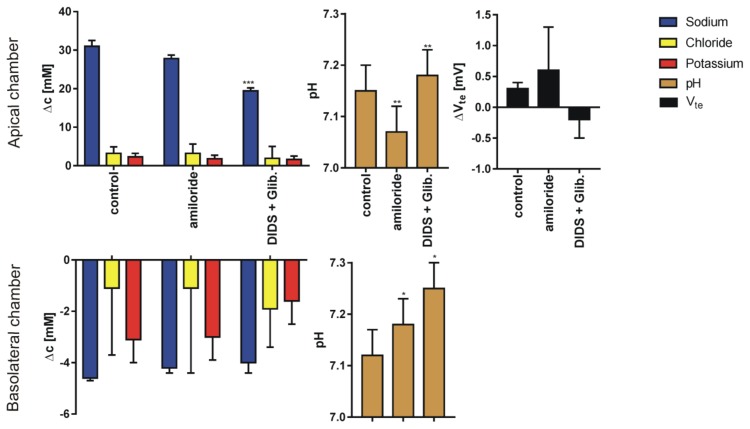
The results (Mean ± SD) of experiments performed in sodium gradient (apical side, KHS2; basolateral side, KHS1) in control conditions and after supplementation of the solution with ion channel blockers. The graph represents: sodium, chloride, potassium transport across the epithelial cell monolayer expressed as the concentration change measured 10 min after appropriate solution flow stop; final pH value and the transepithelial potential difference change (N = 8–10 for Cl^−^, Na^+^ and V_te_, N = 6 for K^+^ and pH, dependent sample t-test, * *p* <0.05, ** *p* <0.01, *** *p* <0.001). Note different scales for apical and basolateral graphs.

**Figure 4 membranes-10-00043-f004:**
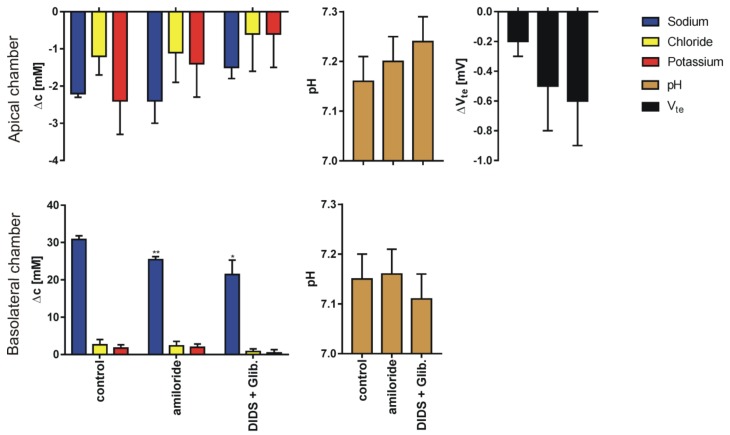
The results (Mean ± SD) of experiments performed in reversed sodium gradient (apical side, KHS1; basolateral side, KHS2) in control conditions and after supplementation of the solution with ion channel blockers. The graph represents: sodium, chloride, potassium transport across epithelial cell monolayer expressed as the concentration change measured 10 min after appropriate solution flow stop; final pH value and the transepithelial potential difference change (N = 7–12 for Cl^−^, Na^+^ and V_te_, N = 6 for K^+^ and pH, dependent sample t-test, * *p* <0.05, ** *p* <0.01, *** *p* <0.001). Note different scales for apical and basolateral graphs.

**Figure 5 membranes-10-00043-f005:**
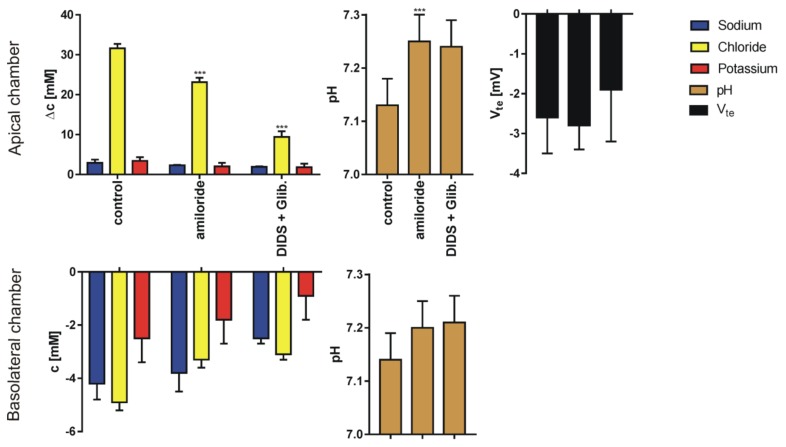
The results (Mean ± SD) of experiments performed in chloride gradient (apical side, KHS3; basolateral side, KHS1) in control conditions and after supplementation of the solution with ion channel blockers. The graph represents: sodium, chloride and potassium transport across epithelial cell monolayer expressed as the concentration change measured 10 min after appropriate solution flow stop; final pH value and the transepithelial potential difference change (N = 15–17 for Cl^-^, Na^+^ and V_te_, N = 5 for K^+^ and pH, dependent sample t-test, * *p* <0.05, ** *p* <0.01, *** *p* <0.001). Note different scales for apical and basolateral graphs.

**Figure 6 membranes-10-00043-f006:**
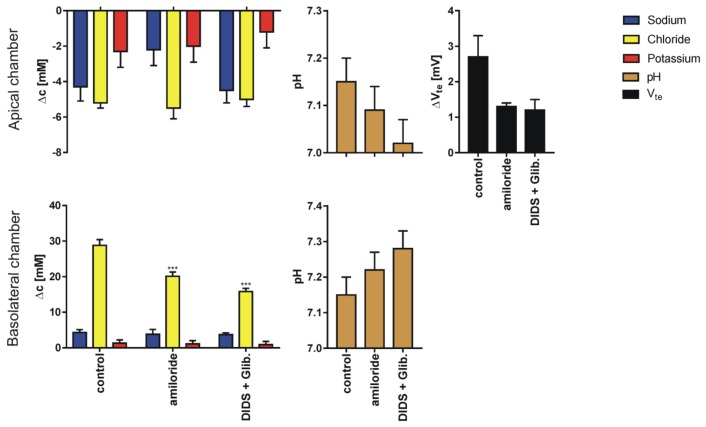
The results (Mean ± SD) of experiments performed in reversed chloride gradient (apical side, KHS1; basolateral side, KHS3) in control conditions and after supplementation of the solution with ion channel blockers. The graph represents: sodium, chloride, potassium transport across epithelial cell monolayer expressed as the concentration change measured 10 min after appropriate solution flow stop; final pH value and the transepithelial potential difference change (N = 10–13 for Cl^−^, Na^+^ and V_te_, N = 5 for K^+^ and pH, dependent sample t-test, * *p* <0.05, ** *p* <0.01, *** *p* <0.001). Note different scales for apical and basolateral graphs.

**Figure 7 membranes-10-00043-f007:**
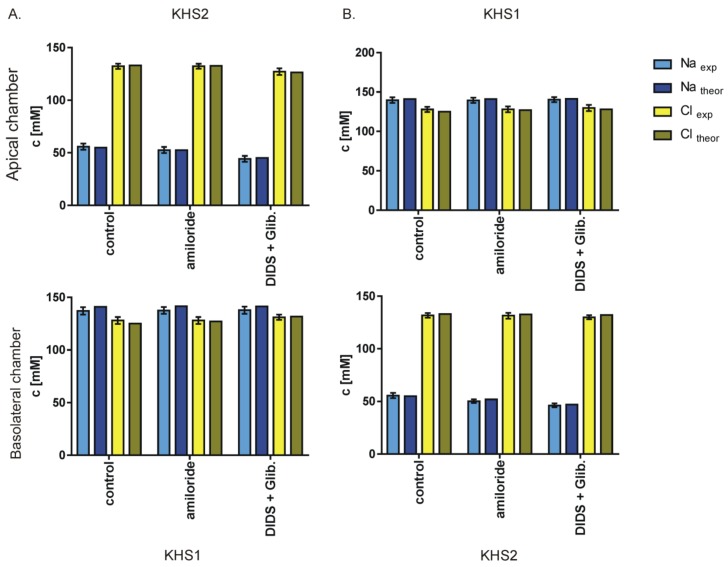
Measured (Mean ± SD) and theoretically predicted concentrations of sodium and chloride ions on both sides of the cell layer in control conditions after blocking particular ion channels present on the apical side of the monolayer. The experiments with sodium gradients and isosmotic flow of *x* fraction of 145 mM sodium chloride across the epithelial cell monolayer are shown.

**Figure 8 membranes-10-00043-f008:**
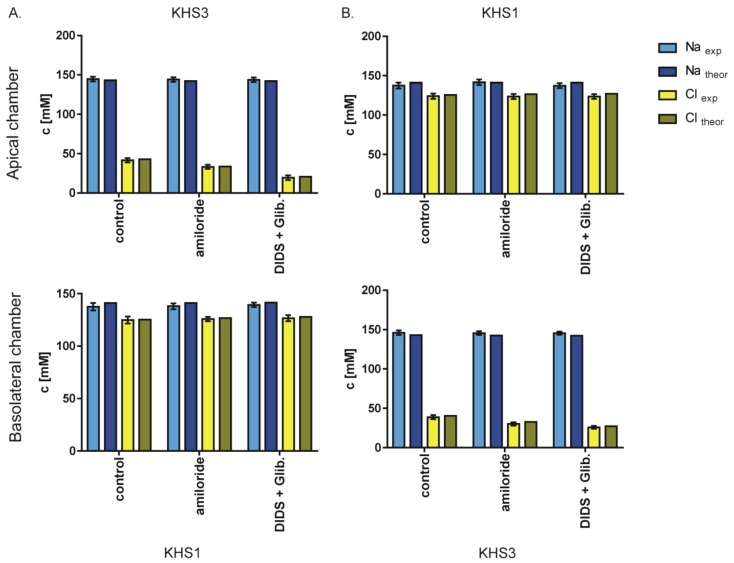
Measured (Mean ± SD) and theoretically predicted concentrations of sodium and chloride ions on both sides of the cell layer in control conditions after blocking particular ion channels present on the apical side of the monolayer. The experiments with sodium gradients and isosmotic flow of *x* fraction of 145 mM sodium chloride across the epithelial cell monolayer are shown.

**Figure 9 membranes-10-00043-f009:**
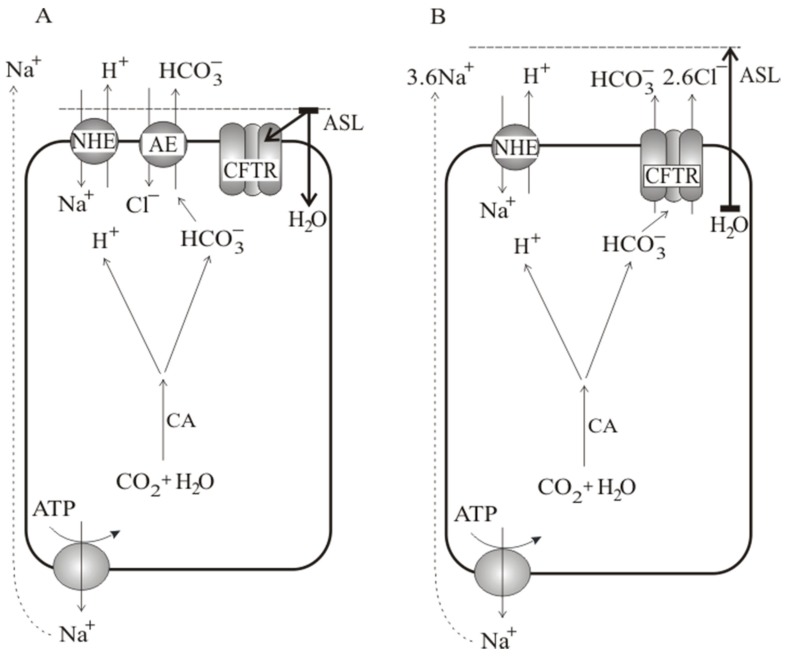
‘The dynamic water balance’ hypothesis: (**A**) water absorption, (**B**) water secretion. During the absorption phase (**A**) two exchangers are responsible for sodium and chloride influx from ASL into the cytoplasm and outward proton and bicarbonate secretion. The lower osmotic pressure of partially dissociated carbonic acid in ASL causes water absorption and depletion of its volume, which increases the concentration of the ASL volume-dependent factor (e.g., ATP) triggering the opening of the CFTR channel and starting the secretion phase (**B**). The secretion of chloride and bicarbonate ions via the CFTR channel is accompanied by the transport of sodium ions via a paracellular pathway. The appearance of net 2.6 NaCl molecules on the apical side leads to osmotic water transport, ASL hydration and dilution of the triggering factor resulting in returning the epithelium to the absorption phase. In the absence of functional CFTR, the bronchial epithelium stays in the absorptive phase leading to Cl/HCO_3_ reaching energetic equilibrium, the action of Na/H exchanger, acidification of ASL and undiluted mucus.

**Table 1 membranes-10-00043-t001:** Ionic composition of media in mM (Π ≈ 289 mOsm/kg H_2_O, pH = 7.4).

Ion	KHS1	KHS2	KHS3
Na^+^	141.8	24.8	141.8
Cl^−^	129.1	129.1	10.0
K^+^	5.9	5.9	5.9
Mg^2+^	1.2	1.2	1.2
Ca^2+^	2.5	2.5	2.5
HCO_3_^−^	24.8	24.8	24.8
H_2_PO_4_^−^	1.2	1.2	1.2
Choline	-	117.0	-
Gluconate	-	-	119.1
Glucose	11.1	11.1	11.1

**Table 2 membranes-10-00043-t002:** Fractions of chamber volume transported in experimental conditions calculated to obtain the best fit between experimental and theoretical predictions.

Buffer in the Chamber		Fraction of Transported Chamber Volume Fluid (%)		
Apical	Basolateral	Control	Amiloride	DIDS+Glibenclamide
KHS2	KHS1	25.9	23.1	17.1
KHS1	KHS2	25.0	22.5	18.6
KHS3	KHS1	24.2	17.5	7.9
KHS1	KHS3	22.5	16.9	14.1

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
