# Peer review of "Measurement of Multi Ion Transport through Human Bronchial Epithelial Cell Line Provides an Insight into the Mechanism of Defective Water Transport in Cystic Fibrosis"

_membranes, 2020, doi:10.3390/membranes10030043_

Round 1

Reviewer 1 Report

Major:

1.       The main issue here is the reviewer/reader is not shown the chamber kinetics. This is critical as it provides a better idea of the mixing in the chamber. What is being measured here is ion concentrations in the apical or basolateral chamber. That is bulk concentration. Not surface concentration.  The volume of a volume or all the cells in the monolayer are trivial to that of the bulk apical and basolateral compartment. Adding to that an unstirred layer that is undefined, a chamber geometry and mixing kinetics that are also undefined and we have values that cannot be interpreted. When the authors measure a 10mM change in chamber concertation. What does that mean? And how much of it came from flow driven by gradients between cells? This is not going to be the same for pH because Na is in the hundreds of millimolar and H+ is in the nanomolar.

2.       Many of the references are not original or directly relevant. This also causes the authors to use obsolete drugs. For example, the authors use a 1996 reference to CFTR block by glybenclamide. There are many more modern and more efficient blockers. Conversely, amiloride was not invented by the Mall group in 2010. So using that as a reference is very misleading and moreover the use of 100 micromolar amiloride which will target ENaC, NHE and virtually all sodium dependent processes is not justified when one can use more NHE specific analogs and more ENaC relevant concentrations or analogs (e.g., 10 micromolar or benzamil).

3.       How many inserts were used when the authors say 269 single determinant? Each insert can obviously be used multiple times and make many single determinants.  What does an actual experiment- raw data look like? How can I judge stability? slope etc...without the raw data.

4.       If amiloride blocks NHE and if Na entry allows H exit why would the medium acidify on the apical side after amiloride? Certainly under open circuit conditions, where there is an apical Na entry, why would this occur? Also under open circuit there is no net current. An epithelium must move both ions- move salts. So how can Na and Cl not match? and glibenclamide+DIDS give a large change of Na but not Cl. Electroneutrality must be maintained. The only way some but not something of this large a discrepancy occur is if ions exhibit different bulk diffusion in aqueous media because of their sizes/shell of hydration. However, in this case it is not relevant to transport by the epithelium. There is a reason why Hans Ussing invented the short circuit techniques- which is to eliminate these issues.

5.       Related to all of the above technical issues is: How can the change in concentration be different for apical vs basolateral sides? If cells volume and amounts of ions are small compared to bulk, then what leaves the apical (reduction) must equal to what arrives in basolateral (addition) and vice versa.

6.       Why are the 16HBE cells a “perfect” cell line? Do you mean ideal or well suited? And again why?

7.       This introduction is a little silly. It is written for people who no nothing of ion and water transport in epithelia. There should be a better introduction regarding CF, water coupling and the bronchiole epithelium itself

Reviewer 2 Report

The study by Zajac et al leads to a model of ion and water transport across an epithelial membrane that is of interest in a very active field of research. For the most part the work appears to be carefully done (but see questions below), and, with revision, should be published.

However, there are some loose ends.

The volume of the chambers is roughly 30 μ The fraction of the chamber volume transported is given to the nearest 0.1%. I was not able to figure out how this was arrived at without the chamber volume being defined to comparable precision. Figures 2-5 have different scales for the two chambers; this makes for nicer looking figures, but, without a warning, would easily lead to misreading the figures. The conductivity of the EnaC channel varies with temperature (Palmer, chapter in Biol Memb Ion Channels, page 425 – 435). No indication of temperature, or, more important, temperature control, is given. Channels generally vary with temperature. It would be useful to have some discussion of the rate of ion transport, and it appears there is enough data in the paper to allow some order of magnitude estimates at least. These in turn could lead to plausible estimates of the density of the channels in the preparation. Specifically:

The conductivity of the EnaC channels is less than 10 pS (Palmer, again). The voltage equivalent of the concentration ratios is always <0.1V (as RT/F(ln(c+Δc/c)), and we may take this as the driving force for each species of ion (as the equation does for the ratio, with the voltage gradient so small). Can enough ions pass through the channels that are included in the model to account for the measured results? If there is at the end a concentration difference of 30 mM x 3 x 10-5L, built up in 600 seconds, if I did my arithmetic correctly, one needs a total of about 1015 ions/second. If this is not the correct interpretation of the data, the authors should be clear about what the correct interpretation is. Given the conductivity, can the authors set a lower limit on the number of channels required in the preparation, and is this number plausible? Since the data have been measured, this should not be a difficult calculation, and would make the paper much more persuasive. As more than one channel is involved, if the conductivities are known, can the ratios of channel numbers be determined?

The AQP (aquaporin) channel has not been referred to at all, although it is apparently involved in the transport of water when there is an osmotic gradient in a somewhat different preparation (Sharma et al, Molec Reprod and Development, 2020). Have the authors ruled out a role for AQP in this preparation? Is the solution always isoosmotic to the point that there is no AQP contribution, os is AQP known to be absent in this preparation? If so, it would be useful to so state.

There were only a couple of errors in English, or perhaps typos, that I found:

line 133: …”what mean that…” (which means that?)

line 193 – 195 The entire sentence does not work.

line 269 “  what    “  (  that?)

On the whole, the English is acceptable.

Overall, this should be a publishable paper, which makes a fairly useful contribution. It would make a stronger contribution if the authors considered the question above.

Author Response

The fraction of the chamber volume transported is given to the nearest 0.1%.

Indeed, we calculated the fraction theoretically to get the best fit between the experiment and theoretical predictions. While the data are surprisingly well reproducible, they should be treated as semi-quantitative before we will be able to measure fluid transport independently. The explanation has been added to Table 2 caption.

Figures 2-5 have different scales …without a warning, would easily lead to misreading the figures.

Thank you. Warning has been added.

No indication of temperature, or, more important, temperature control, is given.

All measurements were performed in room temperature. All the media, cells and equipment were kept in room temperature. There is no heat produced during our experiment. The sentence stating the temperature has been added.

One needs a total of about 1015 ions/second …can the authors set a lower limit on the number of channels required in the preparation, and this number is plausible?

This is truly a smart question; we should have thought about it earlier. First of all, the calculations of Reviewer 2 are correct. Accepting the reviewer comment the following text has been added in the manuscript:

According to Table 2, during 600 seconds of experiment 25% of 30μl of chamber volume is transported across the 1.1cm2 cell layer in the form of 145mM NaCl solution. This is equivalent to the movement of 1015 ions per 1.1cm2 per second or 1.5pA/μm2 which can be delivered by a few open CFTR channels per μm2 or a single VSOR or ORCC channels per few μm2.

The AQP aquaporin channel has not been referred to at all. … in a somewhat different preparation [Sharma et al].

Sharma paper is very interesting indeed. In bronchial epithelium there are AQP 3, 4 and 5. We have not tried to block them to establish whether the transport of water is transcellular or paracellular. The make the point clearer, we add the following sentence:

However, further studies of the mechanism of fluid transport are necessary e.g. whether the water is transported transcellularly via aquaporins [33] or paracellularly.

Reviewer 3 Report

In this manuscript Zajac and coworkers utilize a novel measuring system, designed in house, to measure ion flow through epithelial cells under different ionic gradients.  The apparatus is capable of detecting multiple ions and they are able to quantify the passage of these ions. 

Overall I find the manuscript clear and easy to follow.  The results seam reasonable and the data is adequately presented.

The work culminates in their model explaining the relationship between water and CF when the CFTR chloride channel is inactive.  If find this model compelling.  It would be great if they could add some means of following the passage of water, perhaps using D2O or even T2O if they have the appropriate equipment.  However, as this model is presented as a hypothetical, I do not feel that experiment is necessary for publication at this time.

There are a few places where the english could be strengthened.  However, overall it is clearly written.

Author Response

Thank you for an advice. We are working on a completely new apparatus which will directly measure the volume of fluid transported across the epithelial layer. Preliminary data are very encouraging. It is, however, too early to publish them.

Round 2

Reviewer 1 Report

The changes are appropriate

Author Response

Thank you.

Reviewer 2 Report

The authors appear to have answered essentially all the relevant questions from the first review. Except for minor English usage matters (e.g. line 21 in the abstract “…of the cell monolayer” should be “…across the cell monolayer”), and some commas (phrases must be set off by a pair of commas, and sometimes there is only a comma at the end) this paper would be ready to publish.

However, there is one point that is raised by the new additions (and was actually implied by the questions in the initial review). This concerns the possibility of diffusion across the distances involved (not the membrane transport for which this obviously does not apply, as channels and pumps are needed.) The authors suggest that there must be laminar flow to equilibrate the system, and rule out unstirred layer effects. If I understand the dimensions correctly, the cell layer is 1.1 cm2, and (again assuming I understood correctly) this requires transport across tens of micrometers. If equilibration of ion concentrations takes about 1 s, and the diffusion coefficient is of the order of 10-9 m2 s-1, the time to diffuse 10-5 m (10 µm) is of the order of 2 s. It is possible that diffusion should be accounted for. The other possibility is that I did not understand the paper correctly; dimensions are not labeled in the figures, so this may be the case. if so, the same may be true of other readers. The dimensions and the basis for ruling out a role for diffusion should be clarified.

This said, the paper can be published with just this addition.

Author Response

Reviewer:

The authors appear to have answered essentially all the relevant questions from the first review. Except for minor English usage matters (e.g. line 21 in the abstract “…of the cell monolayer” should be “…across the cell monolayer”), and some commas (phrases must be set off by a pair of commas, and sometimes there is only a comma at the end) this paper would be ready to publish.

Our answer:

Thanks for the kind comment and suggestions. We corrected wording in line 21 and rechecked the use of comes throughout the text.

Reviewer:

However, there is one point that is raised by the new additions (and was actually implied by the questions in the initial review). This concerns the possibility of diffusion across the distances involved (not the membrane transport for which this obviously does not apply, as channels and pumps are needed.) The authors suggest that there must be laminar flow to equilibrate the system, and rule out unstirred layer effects. If I understand the dimensions correctly, the cell layer is 1.1 cm2, and (again assuming I understood correctly) this requires transport across tens of micrometers. If equilibration of ion concentrations takes about 1 s, and the diffusion coefficient is of the order of 10-9 m2 s-1, the time to diffuse 10-5 m (10 µm) is of the order of 2 s. It is possible that diffusion should be accounted for. The other possibility is that I did not understand the paper correctly; dimensions are not labeled in the figures, so this may be the case. if so, the same may be true of other readers. The dimensions and the basis for ruling out a role for diffusion should be clarified.

This said, the paper can be published with just this addition.

Our answer:

In our experiment the medium is exchanged for few minutes. Then the medium flow is stopped. The following questions concerning ion distribution could be raised : i) how do we know that after medium exchange the concentration of ions is uniform in an axis perpendicular to the cell layer (equivalent to unstirred layer effect) and II) parallel to the cell plane.

Answering first question: according to Einstein-Smoluchowski equation time t required for ions to jump the distance x= 3*10-5m is equal to t=x2/2D where diffusion coefficient for potassium and chloride is approximately 2*10-9 m2s-1 thus t=0.225 s. By taking into account that the x value is used for the mean jump distance and 3 jump times are required to get over 99% of uniform distribution we conclude that “less than one second” estimation used in a paper is correct.

The second question is about ions distribution parallel to the plane of cells after exchanging of the medium. We did not assume that the flow is laminar, but we actually observed the dyed medium flowing in a chamber in which the Costar Snapwell porous support was replaced by glass slide. Not only flow was laminar but also the medium was uniformly distributed parallel to the cell plane. Since the exchange of a medium in our chamber took 6 seconds, we used a longer time to assure that there is no previous medium left in a chamber. Accepting the comment and to further clarify this issue, we have amended the manuscript text and rephrased it as follows:

We kept the speed of media exchange at 0.3 ml/min to avoid damage of the cell layer. Stability of the cell layer was monitored by the measurement of the transepithelial resistance. We observed a dyed medium exchange under the microscope using a modified Costar Snapwells where the porous membrane was replaced by a glass slide. We found that the medium flow is laminar. After a few chamber volumes had passed the colour of the medium was uniformly distributed within the chamber excluding possibility of the medium composition difference parallel to the cell plane. In our experimental conditions, the complete medium exchange took 24 seconds. Measurements were made after medium flow stopped. The small thickness of the medium layer excluded the possibility of the unstirred layer effect since the ion concentration differences decay within less than one second. The equilibration of our ISE electrodes took 30-45 seconds - Figure 2.

Reviewer: dimensions are not labeled in the figures.

Our answer:

We agree. We added the information on dimensions to the legend of Figure 1.

Fig.1. Schematic model of apparatus. A – solution inlet and outlet tubes, B – ion-selective and reference electrodes, C – Costar Snapwell insert with epithelial cell monolayer, D – asymmetric body. The diameter of the body is 4 cm. Distance between the cell layer and ion selective electrodes is less than 30µm. The inner diameter of Costar Snapwell insert is 12mm.